# Physical Activity Levels Are Low in Inoperable Lung Cancer: Exploratory Analyses from a Randomised Controlled Trial

**DOI:** 10.3390/jcm8091288

**Published:** 2019-08-23

**Authors:** Lara Edbrooke, Catherine L. Granger, Ross A. Clark, Linda Denehy

**Affiliations:** 1Department of Physiotherapy, The University of Melbourne, Melbourne VIC 3010, Australia; 2Allied Health Department, Peter MacCallum Cancer Centre, Melbourne VIC 3000, Australia; 3Department of Physiotherapy, Royal Melbourne Hospital, Parkville VIC 3050, Australia; 4School of Health and Sport Sciences, University of the Sunshine Coast, Sippy Downs QLD 4556, Australia; 5School of Health Sciences, The University of Melbourne, Melbourne VIC 3010, Australia

**Keywords:** lung cancer, physical activity, exercise, quality of life, symptoms

## Abstract

Background: In inoperable lung cancer, evidence is limited regarding physical activity (PA) and associations with other outcomes. Aims: in the usual care (UC) group of an RCT to (1) explore whether baseline PA was associated with improved follow-up outcomes, (2) identify baseline variables associated with higher follow-up PA and in all RCT participants, to (3) analyse patterns of objectively measured PA, and (4) report on characteristics of those who were able to maintain or increase PA levels. Methods: exploratory analyses of an assessor-blinded RCT. Outcomes, assessed at baseline, nine weeks and six months, included PA (seven-days of accelerometry), six-minute walk distance (6MWD), muscle strength, symptoms, mood and health-related quality of life (HRQoL). Results: 92 participants were randomised, 80 completed baseline accelerometry (39 intervention group (IG), 41 UC), characteristics: mean (SD) age 63.0 (12.3) years, 56% male, 51% stage IV disease. Baseline PA: median (IQR) steps/day 2859.6 (2034.0–3849.2) IG versus 3195.2 (2161.2–4839.0) UC. Associations between baseline PA and six-month outcomes were significant for HRQoL and 6MWD. PA at six months was significantly associated with baseline age, 6MWD and quadriceps strength. Between-group change score (steps/day) mean differences (95% CI) at nine weeks (174.5 (−1504.7 to 1853.7), *p* = 0.84) and six months (574.0 (−1162.3 to 2310.3), *p* = 0.52). Conclusions: further research is required to determine patient subgroups deriving the greatest benefits from PA interventions.

## 1. Introduction

The general health benefits of being physically active are well established. Compared to inactivity, low doses of moderate-intensity exercise are associated with a 22% reduction in all-cause mortality risk in healthy older adults [1]. In general, populations sedentary behaviour is common and when prolonged, is associated with mortality and chronic health conditions, including cardiovascular disease, obesity and type 2 diabetes [2,3,4,5]. Emerging evidence, largely from observational studies, also suggests that increased leisure-time physical activity (PA) is associated with a reduced risk of developing many types of cancer, including lung cancer [6,7,8,9]. In breast, prostate and colorectal cancer populations, increased PA is also associated with risk reductions in cancer mortality and cancer recurrence [10].

US Department of Health and Human Services recommendations for PA in cancer survivors are in line with current recommendations for the healthy population: to perform at least 150 min of moderate-intensity (or 75 min vigorous-intensity) aerobic exercise and two to three resistance-training sessions per week [11,12]. This equates to 30 min of moderate-intensity aerobic exercise five days per week. This goal may not be feasible for some and the individuals unable to meet these PA recommendations should be advised to be as physically active as possible, aiming to make improvements incrementally from baseline values [13]. Previous research has shown that the proportion of people with cancer meeting PA guidelines is low; 18% of UK cancer survivors [14] and 17% with newly diagnosed lung cancer in the US [15]. In Australia, 40% of patients with stage I-IIIB lung cancer scheduled to receive treatment, met current aerobic PA guidelines at the time of diagnosis. Self-reported PA levels declined in this group of patients in the six-months following diagnosis [16].

The benefits of increased PA during or after treatment, largely conducted in populations with breast cancer, include improvements in cardiovascular fitness, muscle strength, symptoms, health-related quality of life (HRQoL), mood and treatment side-effects, including cancer-related fatigue [17,18,19,20]. Increased self-reported PA levels at lung cancer diagnosis are associated with improved survival in people with metastatic disease [21]. In a surgical lung cancer cohort, higher PA levels were associated with improved HRQoL [22]. Improvements in exercise capacity, HRQoL, fatigue and psychological distress following treatment are associated with higher PA levels [22,23]. Numerous methods of assessing PA, by self-report questionnaires or objectively by pedometers and accelerometers, have been previously utilised in lung cancer research [24]. To date, much of the research has relied on patient-reported PA levels [15,21] or has been conducted in the surgical population [22,23,25]. In advanced lung cancer, a cross-sectional study reported moderate correlations between PA levels and HRQoL, dyspnoea, pain and depression [26]. Patients with lung cancer are receptive to the use of simple devices to measure their PA [27] and a wider range of cheaper, validated devices are now available, thus increasing the feasibility of objectively monitoring PA in the clinical setting [28]. For example, most recent smart phones contain activity monitoring sensors that can reasonably accurately determine steps counts and activity bouts. In practice, unfortunately, PA advice is not routinely provided. Clinicians report a number of factors impacting their ability to include discussions regarding PA with their patients, including time constraints, workplace culture, their own knowledge and views on the impact of co-occurring disease or symptoms on PA ability [29] and a lack of clear referral pathways to exercise professionals [30]. However, the ultimate goal in any clinical setting is to provide advice about increasing PA.

In patients with inoperable lung cancer, evidence is limited regarding objectively measured PA levels from the time of commencing active treatment. Similarly, characteristics of patient subgroups who may benefit the most from PA interventions are unknown. The aims of this study in the usual care (UC) group of an RCT were to (1) explore whether baseline PA was associated with improved follow-up outcomes, (2) identify baseline variables associated with higher follow-up PA, and in all RCT participants, to (3) analyse patterns of objectively measured PA, and (4) report on characteristics of those that were able to maintain or increase PA levels.

## 2. Experimental Section

### 2.1. Study Design, Participants and Outcomes

The trial was prospectively registered on the Australian New Zealand Clinical Trials Registry (ACTRN 12614001268639). The trial design has been detailed previously [31,32]. This paper reports exploratory analyses of PA data collected as part of the trial. In summary, the trial was a multi-site assessor-blinded randomised controlled trial with concealed allocation which assessed the effects of home-based multi-disciplinary rehabilitation. Ninety-two participants with inoperable lung cancer were recruited to the trial at the time of commencing active medical treatment. Medical treatment included curative intent chemoradiotherapy, curative intent radiotherapy, palliative chemotherapy, palliative radiotherapy, targeted therapy or immunotherapy. The trial was conducted and reported according to CONSORT [33] and TIDieR [34] guidelines. Ethical approval to conduct the trial was granted on 26 June 2014 by the Peter MacCallum Cancer Centre Human Research Ethics Committee (HREC/14/PMCC/27). All patients gave their informed consent for inclusion prior to participating in the study. Outcomes assessed included functional exercise capacity (six-minute walk distance (6MWD); the primary outcome), physical activity (measured both objectively and by patient self-report using the International Physical Activity Questionnaire (IPAQ)), muscle strength (quadriceps and handgrip), survival and patient-reported outcomes (including HRQoL measured with the Functional Assessment of Cancer Therapy-Lung (FACT-L)), symptom severity and distress (MD Anderson Symptom Inventory-Lung Cancer (MDASI-LC)) and mood (Hospital Anxiety and Depression Scale (HADS)). Outcomes were assessed at baseline, nine weeks and six months, and included a seven-day period of accelerometry following each time point. PA was measured objectively using SenseWear™ accelerometers (APC Cardiovascular Ltd., Crewe, UK) and reported outcomes included average steps per day and PA bouts. Variables relating to PA bouts included the number of daily step bouts of 10+ min duration (derived from minute by minute accelerometer data which included at least 10 consecutive min where any steps were recorded), and the duration and cadence of each of these 10+ min step bouts. Accelerometry data were considered valid and included in the analyses for participants who had met the minimum data requirements of four days, at least eight hours per day wear time [35]. Following baseline testing, participants were randomised (1:1) to receive either usual care (UC) or the intervention (IG). UC consisted of the routine medical and nursing care provided by each of the three trial recruitment sites. At the time the trial was conducted, exercise advice was not routinely provided as part of usual care. The intervention comprised eight weeks of predominantly unsupervised home-based exercise (aerobic and resistance training), behaviour change and symptom management delivered by a physiotherapist and nurse using a combination of home-visits and telephone calls. The intervention was well received by participants [27] and whilst not significantly changing physical function, resulted in significant improvement in patient-reported outcomes (HRQoL, symptom severity and motivation to exercise) at six months [31].

### 2.2. Statistical Analysis

Analyses for the RCT were performed using SPSS v24, R v3.5.0 and Minitab v18 statistical packages. For objectively measured PA, between-group changes between (1) baseline and nine weeks and (2) baseline and six months were analysed using *t*-tests and modified intention-to-treat (mITT). The mITT involved imputation of missing data at nine-week follow-up where participants were known to be alive and had provided data, or at least partial data at nine-weeks. At six months, data were imputed for participants who were known to be alive and had provided at least partial data at nine-weeks. These analyses included baseline characteristics and prior outcome measure scores as covariates. Further details are provided in the trial protocol and statistical analysis plan [31,32].

SPSS v25 was used for all exploratory analyses. Associations between PA, demographics and outcomes at each assessment time (including functional exercise capacity, quadriceps and handgrip muscle strength, mood, symptom severity and distress and HRQoL) were analysed using Spearman’s correlation coefficients. For categorical outcomes with two groups, independent samples *t*-tests or Mann Whitney *U* tests were used to compare PA outcomes between groups, depending on the normality of the data. No data were imputed for these exploratory analyses.

## 3. Results

### 3.1. Baseline Characteristics

Ninety-two participants were randomised (45 IG, 47 UC) to the overall trial [31]. Eighty participants of those recruited into the trial (87%) completed a valid measure of baseline accelerometry (39 IG, 41 UC). Details of participants providing valid objective PA measures at each trial time point are provided in Figure 1.

There were no significant differences in the baseline demographic characteristics of participants who did and did not have valid baseline accelerometry. Table 1 presents baseline demographics and clinical characteristics for the 80 included participants. At baseline, participants were a mean (SD) age of 63.0 (12.3) years, 56% were male and 51% had stage 4 disease. Over half of the participants (59%) rated their baseline performance status as 1 (walking but can only do light work) using the Eastern Co-operative Oncology Group Performance Scale.

### 3.2. Baseline Physical Activity

#### 3.2.1. Physical Activity Levels

At baseline, participants with valid PA data wore the accelerometer for a mean (SD) of 14.3 (3.4) h per day for 6.7 (1.9) days. PA from accelerometry was low in both groups at baseline; median (IQR) steps/day was 2859.6 (2034.0–3849.2) in the IG and 3195.2 (2161.2–4839.0) for UC participants (see Table 2). Thirty-nine percent of participants (31/80) took fewer than 2500 steps per day. In addition, the number of steps accumulated in bouts of 10+ minutes was also low at a median (IQR) of 1.4 (0.8–2.8) and 1.8 (0.8–3.2) 10+ min bouts per day for the IG and UC groups respectively (see Figure 2). Self-reported PA data using the IPAQ indicated that few participants reported meeting PA guidelines at baseline (24% IG versus 29% UC).

#### 3.2.2. Associations Between Physical Activity and Other Variables at Baseline

Associations between objective PA and other variables at baseline are presented for all participants as these measures were completed prior to trial randomisation and commencement of the intervention. Steps per day and the number of 10+ min step bouts per day at baseline were significantly associated with age, baseline functional exercise capacity (6MWD) and several baseline patient-reported measures, including PA levels, HRQoL, depression and symptom distress (Table 3). A higher number of steps and 10+ min step bouts per day at baseline were measured in people who were younger, had higher 6MWD and reported higher PA levels, better HRQoL and lower levels of depression and symptom distress. The association between steps and 10+ min step bouts per day was strong (rho = 0.88, *p* < 0.005).

When assessing the association between PA at baseline and categorical variables, participants who indicated they had met PA guidelines during the previous week on the IPAQ had higher PA for both step and step bout variables. Participants who reported meeting PA guidelines took a median (IQR) of 4919.9 (3195.2 to 7290.4) steps per day compared with 2641.7 (1868.3 to 3753.8) steps per day for those reporting not meeting guidelines (*p* < 0.005). There was also a significant difference in patient-reported weight loss consistent with cachexia (six to twelve months prior to trial recruitment) and the number of 10+ min step bouts per day (median (IQR) of 2.0 (0.9 to 3.2) 10+ min step bouts per day for those without cachexia compared to 1.0 (0.5 to 1.7) 10+ min step bouts per day for those with cachexia, *p* = 0.011). Neither steps per day or number of step bouts were significantly different between groups for any other categorical variable analysed, including sex, living alone, frailty, rural residential location or treatment intent (radical versus palliative).

### 3.3. Change in Physical Activity Levels over Time

There were no significant between-group differences for any PA variable analysed at either follow-up timepoint (9 weeks or 6 months), Table 4 provides further details. Between-group change score mean differences (95% CI) for average steps per day favoured the IG at nine weeks (174.5 (−1504.7 to 1853.7), *p* = 0.84) and six months (574.0 (−1162.3 to 2310.3), *p* = 0.52). 

#### 3.3.1. Associations Between Objective Baseline PA and Follow-up Outcomes

Since the main trial found significant between-group differences favouring the IG for changes in HRQoL and symptom severity from baseline to six-month follow-up, the analyses reported in this section were undertaken using data from the UC group only (*n* = 41). In the UC group, baseline steps per day was not significantly associated with any of the measured follow-up outcomes, including survival at six months, except for the FACT-L trial outcome index (FACT-L TOI) at six-month follow-up (Table 5; *n* = 29, rho = 0.41, *p* = 0.03). The FACT-TOI is a composite outcome of the combined physical and functional wellbeing and lung cancer symptom subscales. Average number of 10+ min of step bouts/day at baseline was significantly associated with six-month follow-up measures of functional exercise capacity (6MWD, *n* = 28, rho 0.407 *p* = 0.03) and HRQoL (*n* = 29, FACT-L scale score rho = 0.42, *p* =0.02; FACT-L TOI rho = 0.51 *p* = 0.005; and FACT-L lung cancer subscale (LCS) rho = 0.37 *p* = 0.05).

#### 3.3.2. Associations between Baseline Variables and Objective PA at Six Months

Steps per day in UC participants (*n* = 27) at six months were significantly associated with age at recruitment (rho = −0.44, *p* = 0.02), with fewer steps being taken by older participants. Both steps per day and number of 10+ min bouts at six months were associated with higher baseline 6MWD (steps per day: rho = 0.42, *p* = 0.03; and number of 10+ min bouts rho = 0.40, *p* = 0.04) and quadriceps force (steps per day: rho = 0.46, *p* = 0.016; and number of 10+ minute bouts rho = 0.52, *p* = 0.006). No other significant associations were found between measured baseline variables and PA at six months.

For the whole group, fifty participants had valid accelerometry data at both baseline and six-month follow-up. Of these, 21 participants (42%, 12 intervention group and 9 usual care group participants) either maintained or increased their average daily steps between the first and final assessment timepoints. These participants had significantly higher 6MWD (mean (SD)) at baseline than participants whose step counts decreased during the study (538.1 (77.3) m versus 476.4 (126.0) m, *p* = 0.038). There were no other significant differences between these groups in terms of demographics or baseline physical function or patient-reported outcomes (PA, HRQoL, symptom severity, symptom distress or mood).

## 4. Discussion

A key finding from the trial PA data is that objectively measured PA was extremely low in people with inoperable lung cancer at baseline (close to commencement of medical treatment). Daily step counts of participants in this trial who wore accelerometers for 14 h per day on average were much lower at baseline than those in previous work from our research group which involved 28 patients with stage I-IIIB LC at diagnosis who took an average of 6120 steps per day [16]. These differences may in part be due to the earlier disease stage and differing treatment modalities of participants in our previous research. Similarly, Bade and colleagues [26] reported higher levels of PA in those with advanced lung cancer, however, their sample comprised participants pre-, during and post-treatment in contrast to our sample of participants who were at the point of commencing medical treatment (either pre or in the two weeks following commencement). Between group changes in PA levels were neither statistically nor clinically [36] significant at either of the trial follow-up time points.

Our participants were less active than participants involved in previous PA research in chronic health conditions, including stable chronic obstructive pulmonary disease (COPD). Over 600 participants over 40 years of age took a median (IQR) of 5112 (333–21191) steps per day [37], and 80 participants involved in a trial of home-based pulmonary rehabilitation who took a mean (SD) of 3836 (2657) steps per day [38]. Activity levels in our trial were also lower than pedometer measures of PA in participants with breast cancer during and following treatment, coronary heart disease and type 2 diabetes [39]. This may be due to multiple factors, including the social isolation and stigma experienced by people with lung cancer [40], the frequent presence of multiple co-morbidities, a symptom burden which is higher than that reported by people with other cancer types [41], competing demands with frequent medical appointments and often, daily hospital outpatient visits for treatment and the poor prognosis often associated with a lung cancer diagnosis.

The findings of our exploratory analyses support those of prior studies which show that higher PA levels were associated with better outcomes [21,22,23,26]. Higher objective PA levels at baseline were associated with being younger and having a higher functional exercise capacity and patient-reported PA, improved HRQoL and lower levels of depression and symptom distress at baseline. Higher activity at six-month follow-up was associated with being younger and having a greater functional exercise capacity and quadriceps muscle strength at commencement of treatment. Whilst statistically significant, these associations were of low to moderate strength (rho = 0.52 being the strongest correlation reported) and are similar to those reported by Bade and colleagues in participants with advanced lung cancer [26]. The strength of these associations is likely reflective of the fact that other factors beyond those measured in the trial influence PA levels in people with lung cancer. The findings at baseline were largely similar, whether assessing PA by the total amount accumulated (steps per day) or patterns of PA (10+ min step bouts). Of interest, only step bouts were significantly associated with baseline symptom severity and participant reports of weight loss consistent with cachexia prior to trial recruitment. This finding highlights the impact that symptom burden and weight loss have on patterns of PA in this population. Whilst both baseline PA variables were associated with follow-up HRQoL, only step bouts at baseline was significantly associated with follow-up levels of functional exercise capacity. This supports previous research in a surgical lung cancer cohort where improvements in exercise capacity, HRQoL, fatigue and psychological distress six-months post-operatively were found to be associated with more time spent in moderate-to-vigorous PA (MVPA) in the month following surgery [23].

Comparison of PA across different studies was impacted by the heterogeneity of variables reported in this area, including steps per day, MVPA, accelerometer counts and self-reported outcomes. PA outcomes chosen for the exploratory analyses reported in this paper included average steps per day and the average number of 10+ min step bouts per day. In this population, it is likely that assessing activity in bouts may be more sensitive to changes than total activity accumulated throughout the day [42]. PA bout data may reflect changes in patterns of PA rather than total accumulated PA, which includes incidental PA, such as walking to go to the bathroom [42]. We chose to analyse step bouts of 10+ min as intervention participants were educated regarding the benefits of PA and were encouraged to walk in bouts of a minimum of 10 min duration throughout the study. We did not include the MVPA variable derived from accelerometry in these exploratory analyses because of the unknown validity of these values in this cohort. Specifically, there is inconsistency in the literature regarding the accuracy of SenseWear^TM^ armbands in measuring energy expenditure (EE). In the healthy population, Santos-Loranzo reported the SenseWear^TM^ armband to over-estimate EE when compared to the gold standard indirect calorimetry [43] (ICCs ranging from 0.66–0.76 during treadmill walking), contrasted by previous findings of under-estimation [44]. In cancer, the accuracy of SenseWear^TM^ armbands to measure EE may be tumour-specific. We observed a lack of agreement between the levels of PA according to steps per day and EE in the RCT [31]. Indirect calorimetry has been used to demonstrate increases in resting EE in advanced lung cancer compared to an age and sex-matched healthy population, prior to commencement of chemotherapy, with no significant changes throughout the course of chemotherapy [45]. It is also worth noting that an acute-phase protein response, defined as a C-reactive protein ≥10 mg L^−1^ was observed in 82% (35/43) of the sample with lung cancer prior to chemotherapy [45]. This is likely to have impacted the validity of the EE calculations derived from the SenseWear^TM^ given that it integrates a temperature sensor in its algorithms.

A limitation of this study is that missing data were not imputed for the exploratory analyses, leading to potential bias, as only participants who were able to complete assessments were included. These participants are likely to represent those with higher function and better patient-reported outcomes, including lower levels of symptom burden. The strengths of this work include the use of accelerometry to objectively measure PA and the inclusion of both total accumulated PA and PA bout variables in the analyses. We would encourage future research to include both total accumulated and bout PA variables in analyses given some of the reported differences in our findings between these two variables. The trial design allowed patterns of PA to be assessed over a six-month period, adding to the limited previous information regarding trajectories of PA for patients with inoperable lung cancer. Importantly, almost half of the participants with data at baseline and six months were able to either maintain or increase their PA levels. This indicates that PA interventions are feasible in at least a large subgroup of this population. In the clinical setting, patients should have access to information regarding PA benefits and those not meeting guidelines who are wanting to become more active should be referred to exercise professionals for individualised advice. Emphasis from clinical staff should be placed on the benefits of incrementally increasing PA rather than achieving 150 min of moderate-intensity exercise per week, which may be an unrealistic target for many. The benefits of being more active have been recently reported in a large sample of older adults where all-cause mortality was found to be significantly reduced in those taking 4400 steps per day compared with inactive participants who took 2700 steps per day [46]. Given the findings that participants with higher PA levels at baseline had higher HRQoL and reduced symptom burden, depression and cachexia at baseline and higher HRQoL at 6 months, further research is required to identify effective interventions to improve PA levels from diagnosis in this population, given the lack of significant change in PA resulting from the intervention of this trial. Our results indicate that performing multiple, extended (10+ min) PA bouts per day is potentially more beneficial than aiming for a set number of steps per day target with no regard for bout duration. However, future research must elucidate whether performing multiple longer walks is more beneficial than performing the same number of steps in repeated short bouts. This is important as the latter may be more feasible in subgroups of this population.

## 5. Conclusions

Physical activity levels are low in people with inoperable lung cancer at the time of commencing medical treatment and do not change significantly in the six months following. Further research is required to determine which patient subgroups may derive the greatest benefits from PA interventions and effective interventions to improve PA from diagnosis.

## Figures and Tables

**Figure 1 jcm-08-01288-f001:**
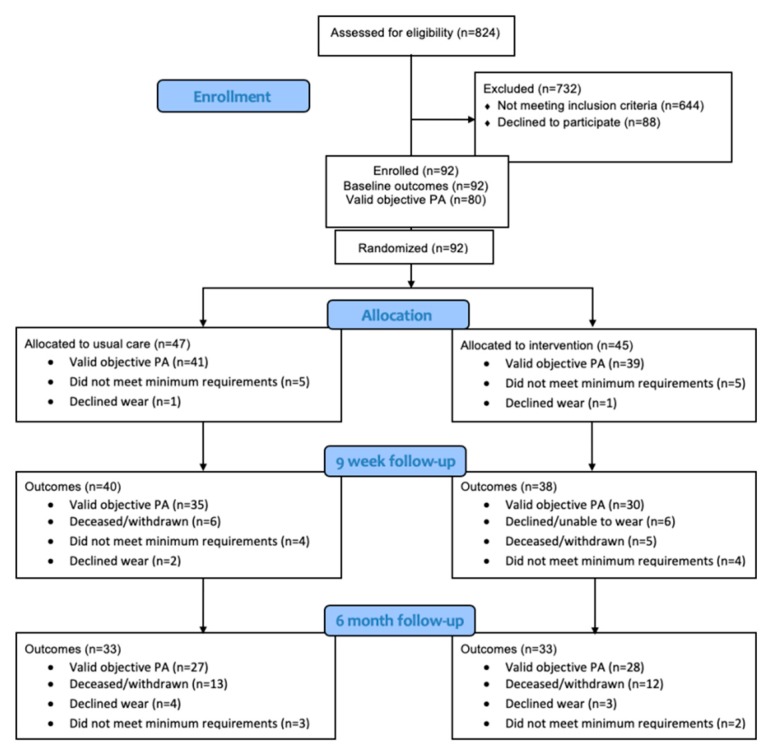
Flow of participants through the trial with valid objective physical activity (PA) measures.

**Figure 2 jcm-08-01288-f002:**
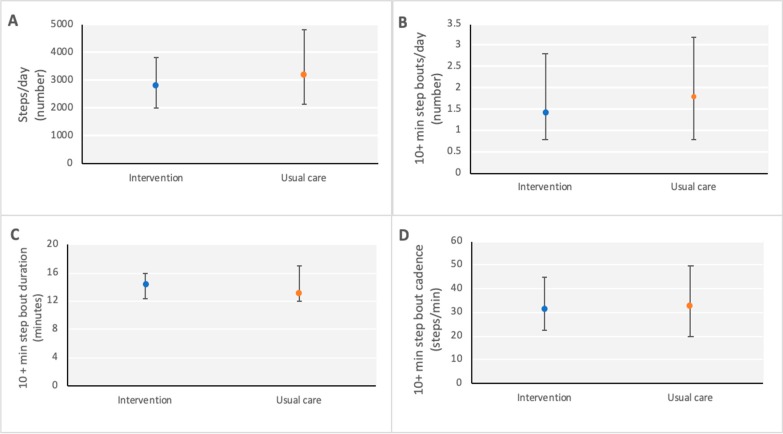
Baseline PA levels: (**A**) Steps per day (number); (**B**) 10+ min step bouts (number of bouts per day); (**C**) 10+ min step bout duration (minutes); (**D**) 10+ min step bout cadence (steps per minute). Values are median (interquartile range).

**Table 1 jcm-08-01288-t001:** Characteristics of participants with valid baseline physical activity data.

	Intervention Group (*n* = 39)	Usual Care (*n* = 41)
Age (years)	64.3 (13.5)	61.9 (11.0)
Sex (male)	19 (49%)	26 (63%)
BMI (kg/m^2^)	26.1 (4.4)	25.8 (4.7)
Time since diagnosis (days)	37.0 (21.5 to 50.0)	41.0 (26.0 to 54.0)
Disease stage		
IA/IB	2 (5%)	1 (2%)
IIIA	9 (23%)	13 (32%)
IIIB	6 (15%)	3 (7%)
IV	18 (46%)	23 (56%)
Recurrence	4 (10%)	1 (2%)
Treatment intent		
Radical	19 (49%)	18 (44%)
Palliative	20 (51%)	23 (56%)
Time from randomisation to commencing treatment * (days)	6.5 (17.6)	4.5 (16.8)
ECOG-PS (patient rated)		
0	11 (28%)	13 (32%)
1	24 (62%)	23 (56%)
2	4 (10%)	5 (12%)
Colinet Co-morbidity score	8.0 (7.0 to 9.0)	8.0 (7.0 to 9.0)
Smoking history		
Never smoker	9 (23%)	6 (15%)
Ex-smoker	22 (56%)	21 (51%)
Current smoker	8 (21%)	14 (34%)

Data are *n* (%), median (IQR) or mean (SD). BMI, body mass index; ECOG-PS, Eastern Cooperative Oncology Group Performance Status. * treatment refers to active medical treatment, including curative intent chemoradiotherapy, curative intent radiotherapy, palliative chemotherapy, palliative radiotherapy, targeted therapy or immunotherapy.

**Table 2 jcm-08-01288-t002:** Baseline physical activity.

	Intervention Group (*n* = 39)	Usual Care (*n* = 41)
Accelerometry		
Steps per day	2859.6 (2034.0 to 3849.2)	3195.2 (2161.3 to 4839.0)
	3382.0 (1972.5)	3801.1 (2414.4)
Number of 10+ min step bouts/day	1.4 (0.8 to 2.8)	1.8 (0.8 to 3.2)
Duration of 10+ min bouts (mins)	14.3 (12.4 to 15.9)	13.1 (12.0 to 17.0)
Cadence of 10+ min bouts (steps/min)	31.6 (22.4 to 44.7)	33.3 (19.8 to 49.4)
Self-reported		
IPAQ (meeting PA guidelines)	10/42 (24%)	12/42 (29%)

Data are *n* (%), median (IQR) or mean (SD). IPAQ = International Physical Activity Questionnaire; PA = physical activity.

**Table 3 jcm-08-01288-t003:** Associations between baseline objective PA, demographics, physical function and patient-reported measures.

Variable	*n*	Steps Per Day	*p*-Value	Number of 10+ Minute Step Bouts	*p*-Value
Age	80	−0.33	0.002	−0.31	0.005
6MWD	80	0.39	<0.005	0.38	0.001
Quadriceps strength	80	0.21	0.069	0.11	0.314
Handgrip strength	80	0.12	0.273	−0.00	0.986
IPAQ	76	0.45	<0.005	0.45	<0.005
FACT-L score	76	0.29	0.011	0.36	0.002
FACT-L TOI	77	0.33	0.004	0.39	0.001
FACT-L LCS	77	0.26	0.025	0.27	0.017
HADS - anxiety	78	−0.04	0.762	−0.02	0.872
HADS - depression	78	−0.24	0.038	−0.29	0.011
MDASI-LC symptom severity	76	−0.22	0.056	−0.31	0.007
MDASI-LC symptom distress	77	−0.31	0.007	−0.32	0.005

Data regarding associations between variables are reported as Spearman’s rho values. FACT-L LCS, Lung Cancer Subscale; FACT-L TOI, Trial Outcome Index (combined physical and functional wellbeing and lung cancer symptom subscales); FACT-L scale, Functional Assessment of Cancer Therapy–Lung; HADS, Hospital Anxiety and Depression Scale; IPAQ; International Physical Activity Questionnaire; MDASI-LC, MD Anderson Symptom Inventory–Lung Cancer (symptom severity subset defined a priori including drowsiness, fatigue, sleep disturbance, shortness of breath and pain); 6MWD, six minute walk distance.

**Table 4 jcm-08-01288-t004:** Within and between-group PA variable change scores at nine-week and six-month follow-ups.

	Within Group Differences				Between Group Differences					
	Intervention Group *		Usual Care *		9-Weeks, *n* = 78 (40 UC, 38 IG) *			6-Months, *n* = 70 (36 UC, 34 IG) *		
	9 Weeks (*n* = 38)	6 Months (*n* = 34)	9 Weeks (*n* = 40)	6 Months (*n* = 36)	Mean Difference (95% CI)	*p*-Value	ES	Mean Difference (95% CI)	*p*-Value	ES
Steps per day	−254.55 (602.31)	361.02 (643.64)	−429.04 (593.82)	−212.96 (634.55)	174.49 (−1504.66 to 1853.65)	0.838	0.05	573.98 (−1162.33 to 2310.29)	0.516	0.15
Number of 10+ min step bouts/day	0.21 (0.54)	0.74 (0.62)	−0.10 (0.47)	−0.37 (0.69)	0.30 (−1.13 to 1.74)	0.680	0.10	1.12 (−0.76 to 2.99)	0.242	0.29
Duration of 10+ min bouts (mins)	−1.75 (1.44)	−1.39 (1.74)	−1.31 (1.56)	−1.98 (1.91)	−0.44 (−4.42 to 3.54)	0.828	0.05	0.59 (−4.61 to 5.78)	0.824	0.06
Cadence of 10+ min bouts (steps/min)	−9.27 (6.71)	−5.22 (5.88)	−1.70 (7.64)	−7.95 (6.52)	−7.57 (−27.29 to 12.15)	0.448	0.17	2.72 (−16.59 to 22.02)	0.780	0.07

* results are from multiple imputation datasets. Mean differences are presented as change scores (9 weeks or 6 months minus baseline) for the intervention group—usual care. *p*-Values were calculated by two-sample independent *t*-tests. Effect size small = 0.2, moderate = 0.5, large = 0.80.

**Table 5 jcm-08-01288-t005:** Associations between baseline PA and follow-up outcomes for usual care participants.

Variable	*n*	Steps per Day	*p*-Value	10+ Minute Step Bouts	*p*-Value
9-week follow-up					
6MWD	35	0.25	0.150	0.27	0.113
Quadriceps strength	33	0.04	0.845	0.10	0.596
Handgrip strength	35	0.05	0.770	−0.02	0.893
FACT-L score	34	0.03	0.857	0.11	0.548
FACT-L TOI	34	0.02	0.898	0.05	0.798
FACT-L LCS	34	−0.03	0.861	0.01	0.959
HADS - anxiety	34	−0.06	0.744	−0.17	0.335
HADS - depression	34	−0.08	0.644	−0.15	0.405
MDASI symptom severity	34	−0.003	0.987	−0.10	0.579
MDASI symptom distress	34	−0.05	0.762	−0.07	0.678
6-month follow-up					
6MWD	28	0.31	0.115	0.41	0.031
Quadriceps strength	28	0.24	0.222	0.25	0.202
Handgrip strength	29	−0.03	0.885	−0.09	0.630
FACT-L score	29	0.34	0.076	0.42	0.024
FACT-L TOI	29	0.41	0.028	0.51	0.005
FACT-L LCS	29	0.23	0.239	0.37	0.049
HADS - anxiety	29	0.06	0.776	0.05	0.796
HADS - depression	29	−0.22	0.257	−0.31	0.101
MDASI symptom severity	29	−0.03	0.893	−0.11	0.587
MDASI symptom distress	29	−0.31	0.098	−0.29	0.134

FACT-L LCS = Lung Cancer Subscale; FACT-L TOI = Trial Outcome Index (combined physical and functional wellbeing and lung cancer symptom subscales); FACT-L scale = Functional Assessment of Cancer Therapy–Lung; HADS = Hospital Anxiety and Depression Scale; MDASI-LC = MD Anderson Symptom Inventory–Lung Cancer (symptom severity subset defined a priori including drowsiness, fatigue, sleep disturbance, shortness of breath and pain); 6MWD, six minute walk distance.

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
