# Peer review of "Physical Activity Levels Are Low in Inoperable Lung Cancer: Exploratory Analyses from a Randomised Controlled Trial"

_jcm, 2019, doi:10.3390/jcm8091288_

Round 1
Reviewer 1 Report
Thank you for this interesting manuscript. I have some general comment, I hope you could add to this manuscript.
In the introduction could you go more into dept on the clinically aspects of an objectively measured PA?
Could an oncologist/clinician use this in daily practice?
What’s practice today? And the evidence
In the discussion could you follow up on the clinically aspects the clinically aspects of an ojectively measured PA?
My point is that the readers you really want to reach are in the clinic meeting these patients every day, they need to recommend PA to their patients and why this is important!
So according to your results (for me) the target group is patients that are old, have a low PA at commencement of treatment. But I think you could get that message out more clearly than it appears in the manuscript now.
Author Response
We thank the reviewers for their time in reviewing our manuscript and for their most useful suggestions. Please find below our responses to the questions raised by the reviewers.
Reviewer 1
Q1 In the introduction could you go more into depth on the clinically aspects of an objectively measured PA? Could an oncologist/clinician use this in daily practice? What’s practice today? And the evidence
R1 Reviewer 1 raises important points and we have modified the manuscript introduction to include the following details in several places in order to address this comment:
Page 2, line 51: This goal may not be feasible for some and those individuals unable to meet these PA guidelines should be advised to be as physically active as possible, aiming to make improvements incrementally from baseline values (Tudor-Locke 2004).
Page 2, line 66: Numerous methods of assessing PA, by self-report questionnaires or objectively by pedometers and accelerometers, have been utilised in lung cancer research previously (Edbrooke et al Respirology).
Page 2, line 71: Patients with lung cancer are receptive to use of simple devices to measure their PA (Edbrooke et al SCC 2019) and a wider range of cheaper, validated devices are now available, thus increasing the feasibility of objectively monitoring PA in the clinical setting. For example, most recent smart phones contain activity monitoring sensors that can reasonably accurately determine steps counts and activity bouts (Althoff et al 2017). In practice, unfortunately PA advice is not routinely provided. Clinicians report a number of factors impacting on their ability to include discussions regarding PA with their patients including; time constraints, workplace culture, their own knowledge and views on the impact of co-occurring disease or symptoms on PA ability (Granger Annals ATS 2016) and a lack of clear referral pathways to exercise professionals (Granger SCC 2017). However, the ultimate goal in any clinical setting is to provide advice about increasing PA.
Q2 In the discussion could you follow up on the clinically aspects the clinically aspects of an objectively measured PA? My point is that the readers you really want to reach are in the clinic meeting these patients every day, they need to recommend PA to their patients and why this is important!
So according to your results (for me) the target group is patients that are old, have a low PA at commencement of treatment. But I think you could get that message out more clearly than it appears in the manuscript now.
R2 In response to the two questions raised above in Q2 by Assessor 1 we provide the following responses:
The low to moderate strength of the associations between PA and outcomes measured in this study make it difficult to definitively define or make specific recommendations regarding which subgroups of patients should be targeted for PA advice. It is in line with previous research that those who were more active at baseline were younger, had higher exercise capacity, had improved health-related quality of life (HRQoL) and reduced depression and symptom distress. Associations between higher PA at 6 months were only significant with baseline HRQoL and exercise capacity. These data were only available for those who were still alive at 6 months and well enough to complete outcome assessment (this information has been added as a study limitation).
This information has been added to the discussion to provide recommendations for health professionals in oncology clinics, line 335:
In the clinical setting patients should have access to information regarding PA benefits and those not meeting guidelines who are wanting to become more active should be referred to exercise professionals for individualised advice. Emphasis from clinical staff should be placed on the benefits of incrementally increasing PA, rather than achieving 150 minutes of moderate-intensity exercise per week which may be an unrealistic target for many. The benefits of being more active have been recently reported in a large sample of older adults where all-cause mortality was found to be significantly reduced in those taking 4400 steps per day compared with inactive participants who took 2700 steps per day (Lee et all JAMA Intern Med 2019).

Reviewer 2 Report
The study was a secondary analysis of physical activity (PA) data from a randomized clinical trial comparing an intervention comprising of 8 weeks of unsupervised home-based aerobic and resistance training with usual care in inoperable lung cancer patients.
The manuscript is generally well written; however, I strongly recommend that the authors carefully review and rewrite the study aims in both the abstract and in the manuscript on Page 2. Much of the analysis is focused on evaluating the relationship of PA and baseline characteristics of all patients included in the study. A second analysis evaluated and compared the change of PA from baseline in patients who received the home-care program and those with usual care at 9 weeks and 6 months follow-up. A third analysis evaluated the relationship between PA measures at 6 months follow-up and baseline variables only in individuals receiving usual care. Finally, a fourth analysis compared baseline characteristics of individuals in the whole group who maintained or increased the PA from baseline to 6 months follow-up compared to those who were observed to have a decrease in their PA. What is the main objective of the study? What are the exploratory analyses? I would suggest you even consider including “…secondary exploratory analyses…” in the study title.
I strongly recommend that the authors avoid such terms as “after commencing treatment” throughout, since it is unclear whether this refers to individuals who participated in the home care program or all participants, given the latter also received general “treatment” as part of their standard care. It would be helpful for the authors to better explain what comprised “usual care”. How many patients in the two groups received chemotherapy and/or radiotherapy?
Page 2, line 74-75: As already mentioned, the study objectives must to be reviewed. The authors looked at the correlation between the 6MWD and other self-reported measures obtained at baseline with objective PA measured at 6 months, rather than what is written: “ whether baseline PA levels were predictive of improved physical function and patient-self reported outcomes at six months”.
Page 2, last paragraph of the Results: how many of the patients who improved rather than had a reduction in their 6MWD received the home-care program?
Page 3, Statistical Analysis, lines 113-114: The analyses for this part of the study should be described as opposed to referring to the previous publications.
Page 3, Statistical Analysis, line 116: “The mITT….”
Page 5, Table 1: Clarify what treatment is being referred to in “time from randomization to commencing treatment”.
Page 5, Heading 3.2: I recommend “Baseline physical activity” rather than “…at treatment commencement”.
Page 6, line 169: I suggest “…for all study participants” rather than “… for the whole group…” I also recommend replacing “relationship” with “correlation” to clarify that you are not performing a regression analysis.
There’s an issue with the page numbering in the manuscript – Discussion Page3 bottom and Page 4: You mention the strengths of the study, what are the study limitations?
Author Response
We thank the reviewers for their time in reviewing our manuscript and for their most useful suggestions. Please find below our responses to the questions raised by the reviewers.
Reviewer 2
Q1. The manuscript is generally well written; however, I strongly recommend that the authors carefully review and rewrite the study aims in both the abstract and in the manuscript on Page 2. Much of the analysis is focused on evaluating the relationship of PA and baseline characteristics of all patients included in the study. A second analysis evaluated and compared the change of PA from baseline in patients who received the home-care program and those with usual care at 9 weeks and 6 months follow-up. A third analysis evaluated the relationship between PA measures at 6 months follow-up and baseline variables only in individuals receiving usual care. Finally, a fourth analysis compared baseline characteristics of individuals in the whole group who maintained or increased the PA from baseline to 6 months follow-up compared to those who were observed to have a decrease in their PA. What is the main objective of the study? What are the exploratory analyses? I would suggest you even consider including “…secondary exploratory analyses…” in the study title.
R1. We have revised the title to describe our analyses as exploratory rather than secondary. It now reads as:
“Physical activity levels are low in inoperable lung cancer: exploratory analyses from a randomised controlled trial.”
We have revised the study aims in the abstract and main manuscript on the basis of Reviewer 2’s comments.
Abstract and the introduction, page 2, line 98:
….in the usual care (UC) group of an RCT to 1) explore whether baseline PA was associated with improved follow-up outcomes; 2) identify baseline variables associated with higher follow-up PA; and in all RCT participants to 3) analyse patterns of objectively measured PA; and 4) report on characteristics of those who were able to maintain or increase PA levels.
Q2. I strongly recommend that the authors avoid such terms as “after commencing treatment” throughout, since it is unclear whether this refers to individuals who participated in the home care program or all participants, given the latter also received general “treatment” as part of their standard care.
R2. We have modified this throughout the manuscript and hope this is now clearer that this refers to the baseline trial assessment which occurred close to the time of commencing medical treatment for lung cancer.
Q3. It would be helpful for the authors to better explain what comprised “usual care”.
R3. P3 line 132, we have added the following information regarding usual care:
UC consisted of the routine medical and nursing care provided by each of the three trial recruitment sites. At the time the trial was conducted exercise advice was not routinely provided as part of usual care.
Q4. How many patients in the two groups received chemotherapy and/or radiotherapy?
R4. All patients received medical treatment for lung cancer as now outlined on p3 line 113 and the footnote of Table 1. As reported in the trial main manuscript (Edbrooke et al Thorax 2019) the groups were balanced in terms of treatment intent (radical/curative versus palliative).
Q5. Page 2, line 74-75: As already mentioned, the study objectives must to be reviewed. The authors looked at the correlation between the 6MWD and other self-reported measures obtained at baseline with objective PA measured at 6 months, rather than what is written: “ whether baseline PA levels were predictive of improved physical function and patient-self reported outcomes at six months”.
R5. This has been amended in the manuscript which now reads
The aims of this study were: in the usual care (UC) group of an RCT to 1) explore whether baseline PA was associated with improved follow-up outcomes; 2) identify baseline variables associated with higher follow-up PA; and in all RCT participants to 3) analyse patterns of objectively measured PA; and 4) report on characteristics of those who were able to maintain or increase PA levels.
Q6. Page 2, last paragraph of the Results: how many of the patients who improved rather than had a reduction in their 6MWD received the home-care program?
R6. In the final paragraph of the results section we have added these details to address this question:
(12 intervention group and 9 usual care group participants)
Q7. Page 3, Statistical Analysis, lines 113-114: The analyses for this part of the study should be described as opposed to referring to the previous publications.
R7. This section was misleading and has been revised. The information provided in paragraph 1 of the statistical analysis section is a summary of the study analyses which are relevant to this manuscript and the references are provided for those wanting further details from the study protocol and statistical analysis plan. This section now reads:
Analyses for the RCT were performed using SPSS v24, R v3.5.0 and Minitab v18 statistical packages. For objectively measured PA, between-group changes between 1) baseline and nine weeks; and 2) baseline and six months were analysed using t-tests and modified intention-to-treat (mITT). The mITT involved imputation of missing data at nine-week follow-up where participants were known to be alive and had provided data, or at least partial data, at nine-weeks. At six months data were imputed for participants who were known to be alive and had provided at least partial data at nine-weeks. These analyses included baseline characteristics and prior outcome measure scores as covariates. Further details are provided in the trial protocol and statistical analysis plan [25,26]
Q8. Page 3, Statistical Analysis, line 116: “The mITT….”
R8. This has been amended in the manuscript.
Q9. Page 5, Table 1: Clarify what treatment is being referred to in “time from randomization to commencing treatment”.
R9. The following details have been added as a footnote to Table 1:
…. treatment refers to active medical treatment including: curative intent chemoradiotherapy, curative intent radiotherapy, palliative chemotherapy, palliative radiotherapy, targeted therapy or immunotherapy.
Q10. Page 5, Heading 3.2: I recommend “Baseline physical activity” rather than “…at treatment commencement”.
R10. This has been amended in the manuscript.
Q11. Page 6, line 169: I suggest “…for all study participants” rather than “… for the whole group…” I also recommend replacing “relationship” with “correlation” to clarify that you are not performing a regression analysis.
R11. This has been amended in the manuscript.
Q12. There’s an issue with the page numbering in the manuscript –
R12. Thank you – we have corrected this in the manuscript.
Q13. Discussion Page3 bottom and Page 4: You mention the strengths of the study, what are the study limitations?
R13. The following information has been added to the Discussion section:
A limitation of this study is that missing data were not imputed for the exploratory analyses performed leading to potential bias as only participants who were able to complete assessments were included. These participants are likely to represent those with higher function and better patient-reported outcomes including lower levels of symptom burden.
